# Correlations among Ultrasonographic, Physicochemical and Sensory Characteristics of Pectoralis Major Muscles in Turkeys Reared in a Sustainable Farming System

**DOI:** 10.3390/ani12010005

**Published:** 2021-12-21

**Authors:** Tomasz Schwarz, Andrzej Węglarz, Krzysztof Andres, Dorota Wojtysiak, Maciej Murawski, Behnaz Ahmadi, Pawel M. Bartlewski, Bahareh Ahmadi

**Affiliations:** 1Department of Animal Genetics, Breeding and Ethology, University of Agriculture in Kraków, 24/28 Mickiewicza Avenue, 30-059 Cracow, Poland; rzschwar@cyf-kr.edu.pl (T.S.); andrzej.weglarz@urk.edu.pl (A.W.); wojtysiakd@wp.pl (D.W.); 2Department of Animal Reproduction, Anatomy and Genomics, University of Agriculture in Kraków, 24/28 Mickiewicza Avenue, 30-059 Cracow, Poland; k.andres@ur.krakow.pl; 3Department of Animal Nutrition, Biotechnology and Fisheries, University of Agriculture in Kraków, 24/28 Mickiewicza Avenue, 30-059 Cracow, Poland; rzmmuraw@cyf-kr.edu.pl; 4Department of Mechanical and Mechatronics Engineering, University of Waterloo, 200 University Avenue W, Waterloo, ON N2L 3G1, Canada; b3ahmadi@uwaterloo.ca; 5Department of Biomedical Sciences, Ontario Veterinary College, University of Guelph, 50 Stone Rd., Guelph, ON N1G 2W1, Canada; pmbart@uoguelph.ca

**Keywords:** turkey, pectoral muscle, meat traits, ultrasonography, image analysis

## Abstract

**Simple Summary:**

One of the challenges of the contemporary poultry industry is to obtain reliable information on meat quality throughout the entire production cycle. Previous studies have shown that computerized analysis of the ultrasonographic images in live birds is a promising method to predict certain characteristics of skeletal muscles. In the present study, the left pectoralis major muscle was scanned just before slaughter in forty-five meat-type turkeys reared in an organic farm. Physicochemical and sensory attributes of the pectoral muscles were determined after slaughter using validated laboratory and analytical methods. There were several significant correlations among ultrasonographic image attributes and physical/sensory characteristics of pectoralis major muscles in the turkeys of the present study, but the moisture content was the only chemical trait associated with ultrasound images. The strongest overall correlation was between pixel heterogeneity obtained in the muscles examined in an oblique plane and aroma. The occurrence and strength of quantitative correlations among ultrasound image characteristics in situ and post-mortem traits of turkeys’ breast are clearly affected by a scanning plane. Computerized analysis of pectoral muscle ultrasonograms provides information on several characteristics that are indicative of meat quality and hence could be potentially used in commercial settings and breed development programs.

**Abstract:**

This study set out to examine associations among echotextural, physicochemical and sensory attributes of the pectoralis major muscles in 17-week-old organic turkeys (B.U.T. Big-6) varying in the amount of wheat and oat grain in daily feed rations (Group C: complete feed only; Group Exp1: 5–30% of wheat and 0–20% of oat; and Group Exp2: 5–50% of wheat and 0–50% of oat; *n* = 15 turkeys/group). Digital ultrasonograms of the left pectoral muscle in four different planes (longitudinal-L, transverse-T, and two oblique planes-O1 and O2) were obtained with a 5.0-MHz linear-array transducer just before slaughter. Mean numerical pixel intensity (MPI) and pixel heterogeneity (MPH) of the muscle parenchyma were computed using the ImageProPlus^®^ analytical software. Ten significant correlations between echotextural attributes and various meat characteristics were recorded in Group C, one in Group Exp1, and eight in Group Exp2. When data were pooled for all birds studied, there were twelve significant correlations (*p* < 0.05); all but one correlation (between MPH and moisture) were for physical and sensory characteristics of meat samples. Computer-assisted analysis is a potential method to determine moisture as well as physical (e.g., coloration) and sensory (e.g., aroma) characteristics of pectoralis major muscles in organic turkeys.

## 1. Introduction

Adequate nutrition is essential for human health and normal development (World Health Organization, https://www.who.int/news-room/facts-in-pictures/detail/nutrition, accessed on 20 October 2021). Turkey meat (the second most widely consumed poultry meat in the world) is characterized by high-protein (up to 28%), low-fat (2% to 5%), and low-cholesterol content, which makes it an important component of a balanced diet [1]. Turkey meat is also a good source of vitamins and minerals, especially B complex vitamins, which prevent anemia and maintain the normal functioning of the cardiovascular and nervous system [2]. In recent years, turkey production and meat processing have turned into a fast-growing branch of poultry industry [3]. Rising meat consumption and customers’ expectations are drawing attention to the issue of meat quality [4]. Across developed countries, poultry producers and researchers try to increase the quality of meat by replacing intensive farming systems with organic farming, placing even greater emphasis on animal well-being [5]. According to the Regulations of the European Community (EC) Commission (no. 889/2008), organic poultry production is defined as the production system that prevents poultry from being reared too quickly; therefore, it mainly utilizes slow-growing poultry strains. In organic poultry farming, animal housing conditions must comply with a high level of standards for animal welfare (e.g., permanent access to open air areas). In addition, animals should only feed on feedstuff produced in accordance with the rules of organic farming. Production systems can impinge on meat quality [6]. Prolonged rearing periods positively affect the chemical composition of breast and leg muscles in poultry, which results in the more desirable aroma and taste of the meat (i.e., better sensory attributes; [7]). Meat from poultry raised in pasture-based or pasture-enriched rearing systems is characterized by higher protein content and low-fat content (both features preferred by consumers) compared with that obtained after intensive rearing [7,8]. Meat from organic chickens has lower pH and water holding capacity, higher shear values, iron content, n-3 polyunsaturated fatty acids and oxidative status, as well as improved color, giving it better overall sensory quality [8,9]. Komprda et al. [10] showed that the content of omega-3 fatty acids in turkey meat can vary widely depending on diet and farming system; the most favorable ratio of omega-6 to omega-3 fats was found in the skinned breast from organic, pasture-raised turkeys [11].

Food production processes include a series of activities and technological operations that require constant control and supervision to ascertain their appropriate progression [12]. One of the challenges facing the contemporary meat industry is to obtain reliable information on meat quality throughout the entire production cycle, which would ultimately provide a guaranteed quality of final products to consumers. To meet this challenge, a fast, accurate and non-invasive technique for determining the physical properties and chemical composition of skeletal muscles in live animals is urgently needed.

Ultrasonography is a non-invasive, inexpensive, widely available, painless (non-sedation-requiring), and real-time imaging and diagnostic technique [13]. Sound waves and their reflections are the basis for displaying ultrasound images [13]. The proportions of reflected and non-reflected waves determine the appearance of ultrasound images composed of numerous pixels ranging in their intensity from 0 (absolute black) to 255 (absolute white; [14]). Ultrasound wave propagation in skeletal muscles depends not only on their composition but also on the structural organization (e.g., orientation of muscle fibers, distribution of connective tissue). However, significant changes in chemical composition may profoundly alter tissue organization and generate multiple acoustic impedance sites that increase backscatter of the echoes (ultrasound waves; [15]). Combining the ultrasound technique with computer-assisted image analysis allows us to extract information from ultrasound images such as the first order textural attributes (mean pixel intensity or numerical values of image brightness elements, and pixel heterogeneity or standard deviation of numerical pixel values; [16,17]). A few experimental and clinical studies have reported a link between echogenicity and proximate chemical composition of various tissues (e.g., human dystrophic muscles [18]; ram testes [19]; and chicken pectoralis major muscle [20]). Image-processing analysis of ultrasonograms could potentially be used to predict intramuscular fat content in live beef cattle; however, the accuracy of this method appears to decline with the increasing fat content [21,22,23]. A personal computer-based image analysis software called USOFT was developed to track the changes in the intramuscular fat content of cattle in the range of 2 to 8% [24], but outside of this range, the software was not able to provide any predictions. An attempt to use real-time ultrasound to predict intramuscular fat content was also made in swine [25] and lambs [26]. Results of those earlier studies showed that computerized analysis of muscle ultrasonograms was a promising tool to predict intramuscular fat percentage in live animals. In the poultry industry, it is possible to estimate carcass cut weights [27] and overall meat content [28,29,30] using in situ ultrasonographic measurements of birds’ skeletal muscles. However, the most advantageous application of ultrasonography would be ability to determine present and future (post-slaughter) physicochemical properties of skeletal muscles. In the most recent study in broiler chickens, Schwarz et al. [20] reported a strong relationship between echotextural characteristics of pectoral muscles and several important meat quality traits such as cutting force, hardness, and chewiness as well as crude fat content. Similar studies do not exist for other poultry species.

Therefore, the primary aim of this study was to examine the echotexture of pectoral muscles (*M. pectoralis superficialis*) for quantitative relationships with their physicochemical and sensory properties in organic turkeys receiving three different types of diet varying in the proportion of cereal (complete feed only or complete feed supplemented with wheat and oats). Wheat has long been considered a primary choice in poultry nutrition, although a proportion of course grains has usually been added to the rations containing wheat [31]. Oats vary considerably in feeding value, due to differences in hull, but several studies and trials have shown that oats are excellent grain for young growing chicks, laying hens and turkeys [32].

## 2. Materials and Methods

### 2.1. Animals, Housing Conditions and Feeding

One-day-old female broad-breasted turkey poults of British United Turkeys (B.U.T.) Big 6 strain (Aviagen Turkeys Ltd., Chester, UK) were randomly assigned to three experimental groups of 100 birds each. Turkey rearing was carried out in compliance with the Regulation of the European Community (EC) Commission no. 889/2008 of 5 September 2008, and the Regulation of the Minister of Agriculture and Rural Development of 18 March 2010, on organic food production (Journal of Laws no. 116, item 975) at the certified organic farm of the Agricultural Production Cooperative “Podhalanka” in Rokiciny Podhalańskie, Poland. A detailed schedule of feeding the turkeys of the present study throughout the rearing period is given in Table 1. The complete organic mixes used in this study consisted of soybean meal, maize, wheat, pea, sunflower meal, triticale, corn gluten, soybean oil, potato protein, monocalcium phosphate, calcium carbonate, mannan oligosaccharide (MOS) extracted from yeast (*Saccharomyces cerevisiae*), sodium chloride, magnesium oxide and sodium bicarbonate in varying proportions. The whole wheat and oats grain came from organic cereal crops. The level of nutrients and metabolizable energy of the mixes are given in Table 2. Throughout the present experiment, the birds had unlimited access to feed and clean water as well as gravel as a source of gastrolytes. They were kept in the floor system on litter and, from the 6th week of life onward, they had access to free range. After 17 weeks of rearing, fifteen birds from each group with a body weight close to the group average were slaughtered. The slaughter was carried out in a commercial abattoir after 10 h of fasting.

### 2.2. Ultrasonography and Echotextural Analyses

Ultrasound scans of the left pectoral muscle were obtained just before slaughter, using an Aloka PS2 ultrasonic scanner (Aloka Ltd., Tokyo, Japan) connected to a hand-held 5.0-MHz linear-array transducer. The muscle was scanned in the transverse, longitudinal and two oblique planes (Figure 1), and still images containing the largest cross-sectional area of the muscle were captured as digital images (DICOM) with a resolution of 640 × 480 pixels. Python algorithms were used to convert original red, green and blue (RGB) images to greyscale ultrasonograms (8 bits). Subsequently, grey-level “stretching” (byte-scale transformation) was used for image normalization. Numerical pixel values were scaled to fill the entire range of display brightness using a formula G_i_ = T(f_i_ − f_min_)/(f_max_ − f_min_), where f_i_ was the original intensity in the range (f_min_, f_max_) and G_i_ was the corresponding scaled intensity in the (0,T) range; T value was equal to 255 [33]. All echotextural analyses were performed using ImageProPlus^®^7.0 analytical software (Media Cybernetics Inc., Rockville, MD, USA). Four identical, non-overlapping spot meters (~33 pixels in diameter) were used to calculate the mean numerical pixel values or pixel intensity (MPI) and pixel heterogeneity (or standard deviations of the mean pixel values-MPH) of each ultrasound image.

### 2.3. Determination of Meat Quality Traits

After cooling at 4 °C for 24 h, the dissected pectoral muscles were weighed and the samples were taken from the central portion of the left pectoralis major of each bird studied to determine chemical composition as well as physicochemical and sensory parameters. Muscle pH was measured using a Matthäus pH meter (Matthäus, Nobitz-Klausa, Germany) with an input electrode normalized for pH 4.0 and 7.0 in accordance with the Polish Standard PN-77/A-82058, and with an automatic adjustment to muscle temperature, at 30 min (pH_30min_) and 24 h (pH_24h_) post-mortem. Meat color was determined using a Minolta CR-310 chromameter (Minolta Co., Ltd., Tokyo, Japan) with a 50-mm diameter measuring head in the CIE Lab system, where the L* parameter corresponds to the degree of lightness (0: black, 100: white), a* and b* represent color components (a* > 0 red, a* < 0 green, b* > 0 yellow, and b* < 0 blue), C* represents chroma (distance from the lightness (L*) axis), and h is the hue angle (expressed in degrees; e.g., 0° is +a*, or red, and 90° is +b*, or yellow; https://sensing.konicaminolta.us/us/blog/understanding-the-cie-lch-color-space/, accessed on 20 October 2021). 

The proximate chemical composition of meat samples (i.e., moisture (oven drying method), total protein (Kjeldahl) and extractable fat (Soxhlet)) was determined using standard laboratory methods [34]. Muscle samples weighing ~120 g were wrapped in aluminum foil and heated in an electric oven at 180 °C until the attainment of an internal temperature of 90 °C. After the heat treatment and subsequent cooling on ice, the thermal weight loss was calculated. Shear (cutting) force was measured using a TA-XT plus texture analyzer (Stable Micro Systems, Godalming, UK) equipped with the Warner-Bratzler shear force attachment [35]. Meat samples (approximately 10 mm × 10 mm × 10 mm) were cut perpendicularly to the visible muscle fibers with a penetration speed of 2 mm/s. At least four cuts were made on each sample and the average values for cutting force calculated. The remaining physical parameters were determined on muscle samples of a similar size (approximately 10 mm × 10 mm × 10 mm) using the same texture analyzer equipped with a 50-mm cylindrical probe. A double compression test was carried out during which the samples were compressed to 40% of their original height along the direction of the muscle fibers; the speed of the head movement was 5 mm/s and the time between successive compressions was 5 s. The following texture parameters were determined using this probe: hardness, springiness, and chewiness. All measurements were taken in duplicate and the values for each variable were calculated automatically. Shear force and Texture Profile Analysis (TPA) values were calculated using the Exponent for Windows ver. 6.1.10.0 (Stable Micro Systems Ltd., Godalming, Surrey, UK; [35]). Breast muscle samples were subjected to sensory analysis by a team of five tasters trained in quality control in accordance with the guidelines described by Baryłko-Pikielna and Matuszewska [36] to assess meat tenderness, cohesiveness, juiciness, aroma (intensity and desirability), and flavor (intensity and desirability); meat flavor was assessed after cooking. Sensory characteristics of meat samples (one raw and one cooked sample per taster) were evaluated using a numerical scale ranging from 1 (poor) to 5 (superior), in increments of 0.5.

### 2.4. Measurements of Muscle Fiber Diameter

The samples (5 g) for microstructural analysis were collected ~30 min after slaughter from the left pectoral major muscle of each carcass. Tissue samples were cut into 1-cm^3^ cubes, frozen in isopentane, and stored at −80 °C until microscopic examinations at a later date. The samples were then mounted on a cryostat holder (SLEE Medical GmbH, Mainz, Germany) with a few drops of tissue freezing medium (Tissue-Tek; Sakura Finetek Europe, Zoeterwoude, The Netherlands). Cross sections (10 μm in thickness and perpendicular to muscle fibers) were cut at −20 °C and stained with hematoxylin and eosin. Finally, all sections were dehydrated in a series of graduated ethyl alcohol, purified in xylene, and immersed in a DPX mounting medium (Fluka, Buchs, Switzerland). At least 300 fibers were counted in each section using a NIKON E600 (Nikon Instech Co. Ltd., Kawasaki, Japan) light microscope. Muscle fiber diameter was quantified using the commercial image analysis system (Multi Scan v. 14.02; Computer Scanning Systems Ltd., Warsaw, Poland).

### 2.5. Statistical Analyses

One-way analysis of variance (ANOVA) and Tukey’s tests were used to compare total weight and physicochemical/sensory variables of pectoral muscles among three groups of turkeys studied (SigmaPlot^®^ 11.0; Systat Software Inc., San Jose, CA, USA). Echotextural differences between groups and scanning planes (transverse vs. longitudinal vs. oblique 1 vs. oblique 2) were analyzed using two-way ANOVA and least significant difference (LSD) tests to compare individual means. The analysis of the moment of Pearson’s product was used to examine correlations between quantitative echotextural attributes and the physicochemical/sensory characteristics. *p* value < 0.05 was considered statistically significant and all results are expressed as mean ± standard error of the mean (SEM).

## 3. Results

### 3.1. Ultrasonographic Assessment

Averaged echotextural variables of the left pectoralis major muscle from organic turkeys studied are shown in Table 3. Mean numerical pixel values obtained in both oblique planes (O1-MPI and O2-MPI) were greater (*p* < 0.05) in Group C compared with Groups Exp1 and Exp2. In addition, mean pixel heterogeneity in oblique plane 1 (O1-MPH) was greater (*p* < 0.05) for Group C than Group Exp2 turkeys but O2-MPH was greater (*p* < 0.05) in Group C compared with Group Exp1. In Group C, O2-MPI was greater (*p* < 0.05) compared with all other scanning planes and it was greater (*p* < 0.05) for L-MPI compared with T-MPI. In Groups Exp1 and Exp2, O2-MPI and L-MPI were greater (*p* < 0.05) compared with T-MPI and O1-MPI. In Group C turkeys, O2-MPH and L-MPH were both greater (*p* < 0.05) than T-MPH and O1-MPH while in Group Exp2, O2-MPH was greater (*p* < 0.05) compared with all other scanning planes, but O1-MPH was less (*p* < 0.05) compared with L-MPH.

### 3.2. Pre-Slaughter Body and Carcass Weights of Turkeys as well as Pectoral Muscle Weight, Proximate Chemical Composition and Fiber Diameter

The fasting weight of birds before slaughter, mean carcass weight and mean weight of pectoralis major muscles were greatest (*p* < 0.05) in control turkeys and they were greater (*p* < 0.05) for Group Exp1 compared with Group Exp2 birds (Table 4). Mean water content of pectoralis major muscles was greatest (*p* < 0.05) for Group Exp2 birds and it was greater (*p* < 0.05) for Group C compared with Group Exp1. There were no differences (*p* > 0.05) among the three experimental groups of organic turkeys in extractable (crude) fat and protein content of pectoralis major muscles. Muscle fiber diameter was greater (*p* < 0.05) in pectoralis major muscles of control turkeys compared with animals allocated to Group Exp2.

### 3.3. Physical and Sensory Characteristics

pH of pectoralis major muscles measured immediately after slaughter was greater (*p* < 0.05) in control turkeys compared with Groups Exp1 and Exp2 (Table 5). L*_0h_ was greater (*p* < 0.05) in Group C than in Group Exp2 and they both were greater (*p* < 0.05) compared with Group Exp1, whereas a*_0h_ and C*_24h_ were greatest for Group Exp1 followed by Group Exp2 and control turkeys (*p* < 0.05). None of the remaining physical properties of pectoralis major muscles differed among the three subsets of organic turkeys (*p* > 0.05).

There were significant differences among the three groups of birds for all sensory characteristics determined in this study except for tenderness and flavor (intensity) of pectoralis major muscle samples (Table 6). Meat cohesiveness, juiciness, aroma (intensity), aroma (desirability) and juiciness were all greater (*p* < 0.05) in Groups Exp1 and Exp2 than in Group C, and flavor (desirability) was greater (*p* < 0.05) in Group Exp2 than in control organic turkeys.

### 3.4. Correlation Analyses

Significant correlations between echotextural variables and physicochemical/sensory attributes of the pectoralis major muscles in organic turkeys studied are summarized in Table 7 and Table 8. When the variables were analyzed for correlations within each of the three groups of birds (Table 7), a total of 19 correlations out of possible 624 (4 scanning planes × 2 echotextural variables × 26 physicochemical or sensory characteristics listed in Table 4, Table 5 and Table 6 × 3 groups) were recorded (3.0%). Ten correlations were recorded in Group C (L-MPI: 1; T-MPI: 1; T-MPH: 1; O1-MPI: 3; O1-MPH: 3; and O2-MPI: 1), one correlation in Group Exp1 (L-MPI), and eight correlations in Group Exp2 (L-MPI: 2; L-MPH: 1; T-MPI: 1; O1-MPI: 2; O2-MPI: 1; and O2-MPH: 1). The strongest linear relationships in Groups C and Exp2 were as follows: between T-MPI and thermal loss (*r* = −0.67, *p* = 0.006) and between L-MPH and moisture (*r* = 0.67, *p* = 0.006), respectively (both moderate correlations according to Guilford and Fruchter, 1978). No correlation between echotextural and chemical/sensory attributes of meat samples was recorded in Group Exp1. No one correlation between the same echotextural variable and physicochemical attribute was seen in all three groups of turkeys nor in Groups C and Exp2.

When data were pooled for all organic turkeys studied, there were twelve moderate overall correlations (12/208 or 5.8% of all possible correlations; *p* < 0.05; Table 8). One correlation was recorded for L-MPH, two for T-MPI, two for O1-MPI, two for O1-MPH, two for O2-MPI, and three for O2-MPH. All but one correlation (between MPH and moisture; r = 0.37, *p* = 0.01) were for physical and sensory characteristics of meat samples. The strongest overall correlation was between O1-MPH and aroma (intensity) (r = −0.41, *p* = 0.005; moderate correlation; [37]).

## 4. Discussion

Replacing a proportion of complete feed mix with wheat and oats resulted in a proportional reduction in body mass, carcass and pectoral muscle weight, and muscle fiber diameter, albeit the latter was statistically significant only between Groups C and Exp2 of 17-week-old turkeys. These changes were accompanied by a decline in muscle moisture in Group Exp1, but by a significant rise in the moisture content in Group Exp2 birds. The plane of nutrition had no apparent effect on intramuscular fat or protein content of pectoral muscles in the organic turkeys studied. Skeletal muscles of vertebrates are characterized by a specific profile of muscle fibers determining their biomechanical function [38]. The number, size and composition of muscle fibers are genetically determined, but they can be changed in response to various extrinsic factors including nutrition [39]. It appears that the addition of large proportions of wheat and oats to daily feed rations had a positive impact on muscle fiber size without altering the crude fat and protein composition of the entire pectoralis major muscle of turkeys. The mechanism of this tropic effects of cereal grains remains to be determined.

Only four of fifteen physical variables analyzed (Table 5) were affected by the change in turkey nutrition, namely pH_0h_, L*_0h_, a*_0h_ and C*_24h_. pH values immediately after slaughter were lower in both treatment groups compared with the control, whereas muscle coloration varied significantly among all three groups of organic turkeys studied, with L*0h/a*0 h and C*24 h being lowest/highest in Groups Exp1 and highest/lowest in Group C. pH has a high influence on water holding capacity (WHC) and the color of meat in different animal species. Low meat pH is often associated with low WHC and pale meat color, whereas high meat pH often causes a dark meat color [40]. These observations are consistent with our present observations on meat coloration immediately after slaughter, but not with the moisture content of the pectoralis major muscles or their color 24 h post-mortem.

Changes in sensory characteristics of meat samples recorded in the present study were remarkably consistent. Apart from meat tenderness and flavor (intensity) that did not vary among the three groups of birds, and flavor (desirability), which only differed between Groups C and Exp2, all other attributes were greater for both experimental groups compared with a control subset of organic turkeys. Meat flavor can be affected by many factors including, but not limited to, feed used, aging after slaughter and/or cooking method [41]. Clearly, an increased proportion of cereal in organic turkeys’ nutrition had a positive impact on meat characteristics assessed by the expert sensory panel.

Mean pixel intensity and heterogeneity of the pectoralis major muscle in turkeys of the present study was generally greater for images obtained in L and O2 compared with T and O1 planes. The pectoralis muscle attaches to the humerus of the wing at the deltopectoral crest, its main portion (*sternobrachialis*) originates from an enlarged sternal keel, with more anterior fibers arising from the furcular, and a much smaller portion (*thoracobrachialis*) originates dorsally from ribs [42,43]. Therefore, the course of the fibres of the pectoralis major muscle was largely perpendicular to the probe held in L or O2 position and more parallel when the probe was held in T and O1 planes (Figure 2). Due to reflections of ultrasound waves by the perimysial connective tissue of the fibers, which is moderately echogenic, the muscle has a “linear” and less echogenic appearance in the longitudinal plane (parallel to the muscle fibers) and “speckled” or more echogenic appearance in the transverse plane (perpendicular to muscle fibers; [14]). Interestingly, the only MPI and MPH values that differed among the three groups of birds were obtained in O2 and O1 scanning planes. All echotextural characteristics of the pectoralis major muscles in the turkeys of the present study were numerically higher in Group C compared with both Exp groups, and the differences for O1-MPI and O2-MPI were statistically significant. The reason for this is difficult to explain. Although O2-MPI and O2-MPH values were consistently highest among the four scanning planes, O1-MPI and O1-MPH showed either intermediate or lowest values. In healthy humans, an increase in skeletal muscle echointensity usually results from loss in muscle mass (quantitative change), declining number and size of muscle fibers [44] and/or age-related accumulation of fat and fibrous tissue (qualitative change; [45]). The diameter of muscle fibers is higher is animals fed high-protein feed compared with those receiving low-protein diets, while the concurrent change in density of muscle fibers has shown the opposite trend [46,47]. Skeletal muscles with larger and less densely distributed muscle fibers generally exhibit higher pixel intensity of skeletal muscles [48]. In the turkeys of the present study, lower pixel intensity of pectoralis major muscles in experimental groups of turkeys were associated with lower (numerically or significantly) body weight and muscle fiber size, but there were no differences in crude fat or protein content of the muscles among the three groups of birds studied. Therefore, muscle fiber diameter may be one of the most important factors determining echotextural properties of pectoralis major muscles in turkeys.

The analyses of quantitative correlations among echotextural and sensory/physicochemical characteristics of the pectoral muscle revealed different numbers of significant correlations within each group, ranging from one correlation in Group Exp1 and 8 to 10 correlations in Groups C and Exp2. This is puzzling and a reason for such an uneven distribution of significant correlations remains unknown. A large number of significant correlations (*n* = 12) were recorded with the data pooled for all turkeys studied. Most correlations were with meat physical properties, luminosity, color, and sensory characteristics (e.g., taste and aroma). The only chemical constituent that was significantly correlated with echotextural variables, within the groups and for all animals studied, was moisture. The latter contrasts with earlier ultrasonographic studies of the pectoral muscles in broiler chickens [20] and different mammalian organs/tissues [19,49] where significant correlations were consistently seen between first order echotextural variables and fat/protein content. Although the fact that turkey meat is less fatty than that in other livestock species may explain a lack of correlations with crude fat content, a reason for a lack of significant correlations with crude protein remains unknown.

Higher numerical pixel values of ultrasonograms reflect the occurrence of harder and more compact tissue fragments [50,51]. Therefore, a positive correlation between MPI and cutting force seen for T and O2 scanning planes (Groups C and Exp2, respectively) seems logical. Usually, ultrasound image echointensity is adversely related to anechoic water content, or moisture, but in the present study, such correlations were restricted to pixel heterogeneity of pectoral muscles (MPH). A significant negative correlation was also noted between MPI/MPH and thermal loss in the control group of turkeys. The pectoral muscle in this group was characterized by highest fat content and largest muscle fiber diameter, causing lowest juiciness; this may explain the highest MPI/MPH values and their association with the thermal loss.

## 5. Conclusions

To summarize, the addition of large proportions of cereal (wheat and/or oats) to daily feed rations altered several physical and sensory characteristics of the pectoralis major muscle in organic turkeys but had less effect on their chemical composition (confined to moisture content). There exist several significant correlations among echotextural attributes and physical/sensory characteristics of pectoralis major muscles, but not to their proximate chemical composition (restricted to moisture) in animals fed complete feed and organic turkeys receiving very high proportion of cereal grains in their diet. The reasons for a lack of correlations with muscle chemical constituents remain unknown.

## Figures and Tables

**Figure 1 animals-12-00005-f001:**
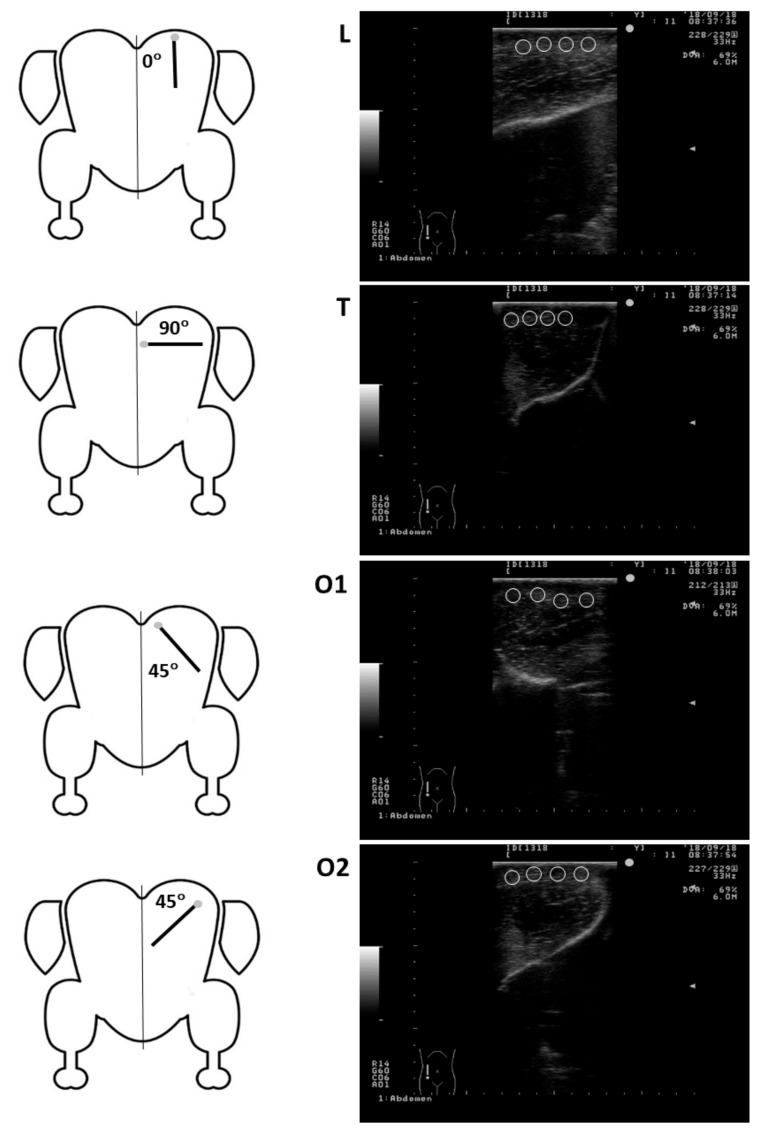
Diagrammatic representation of scanning planes (**left** panels; L-longitudinal, T-transverse; O1 and O2-oblique) and corresponding ultrasonographic images of pectoral muscles (**right** panels) in ultrasonographically examined organic turkeys. The angels are between the midline axis (sternum and keel bone) and the casing of a transducer placed above the left pectoralis major muscle. Light grey dots in both sets of panels show direction of the transducer probe applied to the bird’s skin. White hollow circles in ultrasonograms illustrate potential placement of spot meters used for echotextural analyses of muscle tissue.

**Figure 2 animals-12-00005-f002:**
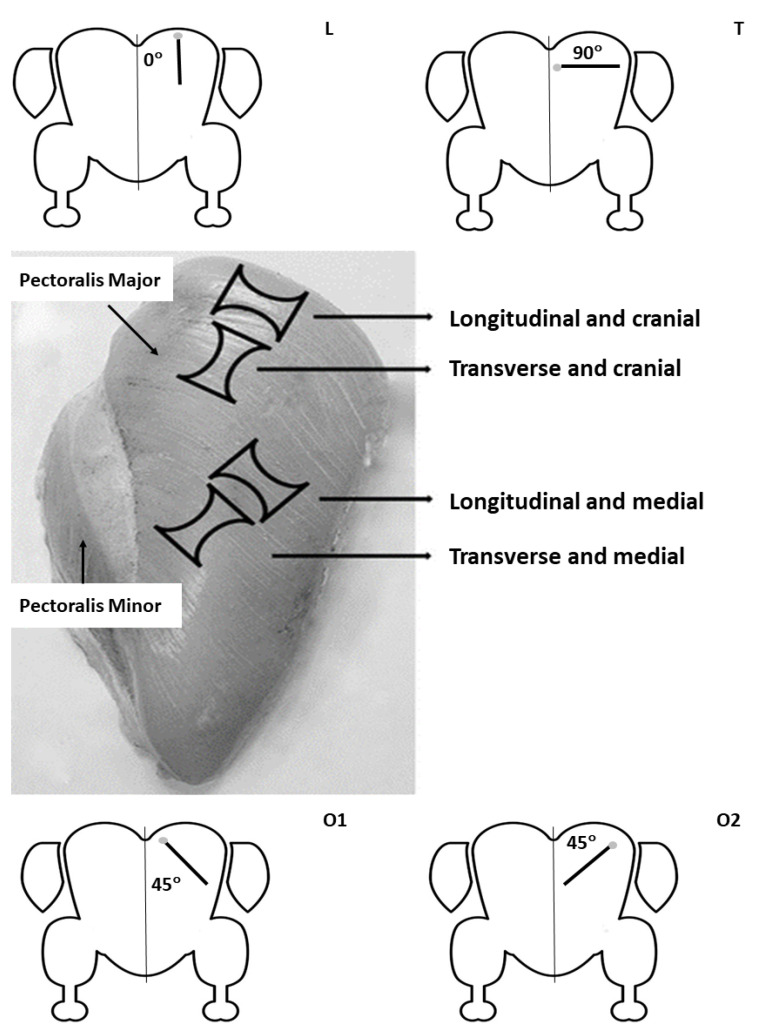
A figure illustrating the course of muscle fibers within the cranial and medial segments of the pectoralis major muscle of turkeys in relation to scanning planes or a position of the shaft of a transducer during ultrasonographic examinations (L-longitudinal, T-transverse, and O1 and O2-oblique). The angels are between the midline axis (sternum and keel bone) and the casing of a transducer placed above the left pectoralis major muscle. Light grey dots show direction of the transducer probe applied to the bird’s skin.

**Table 1 animals-12-00005-t001:** Feeding schedule of control (C) and experimental (Exp) groups of organic turkeys.

Age (weeks)/Group	Group C	Group Exp1	Group Exp2
0 to 4	Complete feed I	Complete feed I (95%) + wheat (5%)	Complete feed I (95%) + wheat (5%)
5 to 8	Complete feed II	Complete feed II (90%) + wheat (7%) + oat (3%)	Complete feed II (84%) + wheat (10%) + oat (6%)
9 to 12	Complete feed III	Complete feed III (80%) + wheat (14%) + oat (6%)	Complete feed III (63%) + wheat (25%) + oat (12%)
13 to 14	Complete feed IV	Complete feed IV (70%) + wheat (20%) + oat (10%)	Complete feed IV (40%) + wheat (35%) + oat (25%)
15 to 17	Complete feed IV	Complete feed IV (50%) + wheat (30%) + oat (20%)	Wheat (50%) + oat (50%)

**Table 2 animals-12-00005-t002:** Chemical composition and metabolizable energy of the feed and grains used throughout the rearing period of organic turkeys.

Item	Dry Matter (%)	Crude Ash (%)	Crude Protein (%)	Crude Fat (%)	Crude Fiber (%)	Metabolizable Energy (kcal/kg)
Complete feed I	91.56	12.12	30.38	4.14	3.77	2747.9
Complete feed II	90.55	7.25	24.88	5.18	4.91	2876.4
Complete feed III	91.61	8.50	24.00	6.10	4.37	2843.5
Complete feed IV	91.07	7.86	22.81	5.58	4.22	2822.8
Oat grain	89.92	2.64	10.50	4.01	12.08	2543.7
Wheat grain	87.22	1.94	12.81	2.09	2.89	3037.4

**Table 3 animals-12-00005-t003:** Echotextural attributes of pectoralis major muscles in organic turkeys receiving three different diets.

Variable/Group	Group C	Group Exp1	Group Exp2
L-MPI	52.6 ± 1.9 ^b^	47.9 ± 2.6 ^a^	48.3 ± 1.9 ^a^
T-MPI	42.0 ± 3.0 ^c^	38.8 ± 2.1 ^b^	36.4 ± 1.8 ^b^
O1-MPI	47.6 ± 2.4 ^bcA^	38.7 ± 1.8 ^bB^	35.5 ± 1.6 ^bB^
O2-MPI	56.6 ± 2.8 ^aA^	49.5 ± 2.6 ^aB^	48.3 ± 1.5 ^aB^
L-MPH	19.3 ± 0.5 ^a^	17.5 ± 0.4	18.7 ± 0.5 ^b^
T-MPH	17.1 ± 0.6 ^b^	17.5 ± 0.5	17.6 ± 0.6 ^bc^
O1-MPH	18.5 ± 0.5 ^bA^	17.3 ± 0.5 ^AB^	16.1 ± 0.6 ^cB^
O2-MPH	21.5 ± 0.8 ^aA^	19.3 ± 0.7 ^B^	20.3 ± 0.6 ^aAB^

Groups C (control), Exp1 and Exp2; *n* = 15 birds/group; L: longitudinal plane; T: transverse plane; O1 and O2-oblique planes; MPI: mean numerical pixel values (pixel intensity); and MPH: mean pixel heterogeneity (standard deviation of mean numerical pixel values). Means denoted by different letters are significantly different (*p* < 0.05): ^a–c^ between scanning planes for each echotextural variable (MPI or MPH); ^A–B^ between groups.

**Table 4 animals-12-00005-t004:** Mean (*±* SEM) body and carcass weight, proximate chemical composition, and muscle fiber diameter of pectoralis major muscles in organic turkeys receiving three different diets.

**Variable/Group**	**Group C**	**Group Exp1**	**Group Exp2**
Body weight before slaughter (kg)	10.7 ± 0.06 ^a^	10.3 ± 0.07 ^b^	9.8 ± 0.06 ^c^
Mean carcass weight (kg)	8.8 ± 0.04 ^a^	8.1 ± 0.06 ^b^	7.8 ± 0.04 ^c^
Mean weight of pectoralis major muscles (g)	1369 ± 22 ^a^	1239 ± 20 ^b^	1114 ± 13 ^c^
Moisture (%)	73.1 ± 0.1 ^b^	72.1 ± 0.2 ^c^	74.2 ± 0.3 ^a^
Protein (%)	24.35 ± 0.29	24.30 ± 0.31	23.86 ± 0.16
Crude fat (%)	2.03 ± 0.22	1.19 ± 0.16	1.79 ± 0.23
Muscle fiber diameter (µm)	75.5 ± 2.0 ^a^	72.4 ± 1.7 ^ab^	68.9 ± 0.9 ^b^

Groups C (control), Exp1 and Exp2; *n* = 15 birds/group; ^a–c^ Within rows, mean values denoted by different letters are significantly different (*p* < 0.05).

**Table 5 animals-12-00005-t005:** Physical properties of pectoralis major muscles in organic turkeys receiving three different diets.

Variable/Group	Group C	Group Exp1	Group Exp2
pH_0h_	6.48 ± 0.05 ^b^	6.07 ± 0.06 ^a^	6.06 ± 0.07 ^a^
L*_0h_	61.6 ± 0.4 ^a^	53.0 ± 0.9 ^c^	56.0 ± 1.0 ^b^
a*_0h_	6.5 ± 0.08 ^c^	9.2 ± 0.4 ^a^	8.1 ± 0.4 ^b^
b*_0h_	−1.5 ± 0.2	−1.1 ± 0.2	−1.8 ± 0.3
pH_24h_	5.61 ± 0.02	5.61 ± 0.02	5.65 ± 0.03
L*_24h_	57.6 ± 0.5	57.1 ± 0.6	56.8 ± 0.7
a*_24h_	10.2 ± 0.2	10.3 ± 0.2	10.9 ± 0.4
b*_24h_	1.6 ± 0.4	1.5 ± 0.2	1.2 ± 0.3
C*_24h_	6.8 ± 0.1 ^c^	9.3 ± 0.4 ^a^	8.4 ± 0.3 ^b^
h_24h_	−1.14 ± 0.19	−1.23 ± 0.20	−1.14 ± 0.19
Hardness (N)	100.8 ± 0.9	110.9 ± 10.2	118.0 ± 11.7
Springiness (N)	0.50 ± 0.02	0.57 ± 0.02	0.52 ± 0.01
Chewiness (N)	22.2 ± 2.4	28.3 ± 3.3	27.8 ± 3.2
Cutting force (N)	18.0 ± 0.8	20.1 ± 3.5	23.7 ± 1.6
Thermal loss (%)	28.4 ± 0.7	26.6 ± 0.7	28.3 ± 0.8

Groups C (control), Exp1 and Exp2; *n* = 15 birds/group; ^a–c^ Within rows, mean values denoted by different letters are significantly different (*p* < 0.05). L* parameter corresponds to the degree of lightness (0: black, 100: white), a* and b* represent color components (a* > 0 red, a* < 0 green, b* > 0 yellow, and b* < 0 blue), C* represents chroma (distance from the lightness (L*) axis).

**Table 6 animals-12-00005-t006:** Sensory characteristics of pectoralis major muscles in organic turkeys receiving three different diets.

Variable/Group	Group C	Group Exp1	Group Exp2
Tenderness	4.44 ± 0.03	4.50 ± 0.04	4.53 ± 0.04
Cohesiveness	4.46 ± 0.04 ^b^	4.65 ± 0.003 ^a^	4.64 ± 0.03 ^a^
Juiciness	4.36 ± 0.04 ^b^	4.51 ± 0.04 ^a^	4.55 ± 0.04 ^a^
Aroma (intensity)	4.40 ± 0.04 ^b^	4.55 ± 0.02 ^a^	4.61 ± 0.03 ^a^
Aroma (desirability)	4.38 ± 0.03 ^b^	4.53 ± 0.03 ^a^	4.60 ± 0.03 ^a^
Flavor (intensity)	4.49 ± 0.03	4.48 ± 0.03	4.57 ± 0.03
Flavor (desirability)	4.40 ± 0.02 ^b^	4.48 ± 0.04 ^ab^	4.60 ± 0.03 ^a^

Groups C (control), Exp1 and Exp2; *n* = 15 birds/group; ^a–c^ Within rows, mean values denoted by different letters are significantly different (*p* < 0.05). In Group C, and flavor (desirability) was greater (*p* < 0.05) in Group Exp2 than in control organic turkeys.

**Table 7 animals-12-00005-t007:** Summary of significant correlations among echotextural variables and sensory/physicochemical properties of pectoralis major muscles recorded in organic turkeys allotted to three different nutritional groups.

Input Variable (x)	Output Variable (y)	*r*	*p* Value	Regression Equation
Group C				
L-MPI	L*_0h_	0.54	0.04	y = 54.69 + 0.13x
T-MPI	Thermal loss	−0.67	0.006	y = 34.54 − 0.15x
T-MPH	Cutting force (N)	0.59	0.03	y = 5.45 + 0.73x
O1-MPI	C*_24h_	0.59	0.02	y = 5.60 + 0.02x
O1-MPI	Protein	−0.57	0.03	y = 27.67 − 0.07x
O1-MPI	Flavor (intensity)	0.51	0.05	y = 4.14 + 0.007x
O1-MPH	Thermal loss	−0.57	0.02	y = 42.60 − 0.77x
O1-MPH	h_24h_	0.54	0.04	y = −4.98 + 0.21x
O1-MPH	Moisture	0.51	0.05	y = 71.00 + 0.11x
O2-MPI	Cutting force (N)	0.66	0.07	y = 6.86 + 0.20x
Group Exp1				
L-MPI	b*_0h_	−0.57	0.03	y = 0.91 − 0.04x
Group Exp2				
L-MPI	Hardness	−0.55	0.03	y = 0.56 − 0.002x
L-MPI	L*_0h_	−0.51	0.05	y = 68.96 − 0.27x
L-MPH	Moisture	0.67	0.006	y = 66.24 + 0.42x
T-MPI	Cutting force (N)	0.58	0.02	y = 0.53 + 0.05x
O1-MPI	L*_0h_	−0.57	0.03	y = 68.34 − 0.35x
O1-MPI	a*_0h_	0.63	0.01	y = 3.20 + 0.14x
O2-MPI	Flavor (desirability)	0.57	0.03	y = 3.98 + 0.01x
O2-MPH	Cohesiveness	−0.53	0.04	y = 5.24 − 0.03x

Groups C (control), Exp1 and Exp2; *n* = 15 birds/group; *r*: coefficient of correlation; L: longitudinal plane; T: transverse plane; O1 and O2-oblique planes; MPI: mean numerical pixel values (pixel intensity); and MPH: mean pixel heterogeneity (standard deviation of mean numerical pixel values). L* parameter corresponds to the degree of lightness (0: black, 100: white), a* and b* represent color components (a* > 0 red, a* < 0 green, b* > 0 yellow, and b* < 0 blue), C* represents chroma (distance from the lightness (L*) axis).

**Table 8 animals-12-00005-t008:** Summary of significant correlations among echotextural variables and sensory/physicochemical characteristics of pectoralis major muscles recorded in organic turkeys; data were pooled for all birds studied.

Input Variable (x)	Output Variable (y)	*r*	*p* Value	Regression Equation
L-MPH	Moisture	0.37	0.01	y = 68.61 + 0.22x
T-MPI	L*_24h_	0.29	0.05	y = 54.49 + 0.07x
T-MPI	Thermal loss	−0.30	0.04	y = 31.39 − 0.09x
O1-MPI	Aroma (desirability)	−0.30	0.05	y = 4.70 − 0.005x
O1-MPI	C*_24h_	−0.34	0.02	y = 10.52 − 0.06x
O1-MPH	Cohesiveness	−0.36	0.02	y = 5.01 − 0.02x
O1-MPH	Aroma (intensity)	−0.41	0.005	y = 5.00 − 0.03x
O2-MPI	L*_0h_	0.30	0.05	y = 49.39 + 0.14x
O2-MPI	C*_24h_	−0.30	0.05	y = 10.62 − 0.05x
O2-MPH	C*_24h_	−0.31	0.04	y = 11.66 − 0.17x
O2-MPH	L*_0h_	0.33	0.03	y = 45.74 + 0.54x
O2-MPH	Aroma (intensity)	−0.31	0.04	y = 4.85 − 0.02x

*r*: coefficient of correlation; L: longitudinal plane; T: transverse plane; O1 and O2: oblique planes; MPI: mean numerical pixel values (pixel intensity); and MPH: mean pixel heterogeneity (standard deviation of mean numerical pixel values). L* parameter corresponds to the degree of lightness (0: black, 100: white), C* represents chroma (distance from the lightness (L*) axis).

## Data Availability

None of the data were deposited in an official repository. All data are available upon reasonable requests.

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
