# Peer review of "Correlations among Ultrasonographic, Physicochemical and Sensory Characteristics of Pectoralis Major Muscles in Turkeys Reared in a Sustainable Farming System"

_animals, 2021, doi:10.3390/ani12010005_

Round 1
Reviewer 1 Report
Please find a list of remarks in the attached file

Author Response
Dear Mam/Sir,
Thank you so much for your constructive criticism.
We've updated our manuscript to reflect your suggestions, as mentioned in the attached.
Regards,
Bahareh Ahmadi DVM, MSc, PhD candidate (Biomedical Sciences)

Reviewer 2 Report
Dear Authors,
The article is interesting and well-developed. Correlations between echotextural, physicochemical and sensory attributes of the pectoralis muscles in turkeys were assessed, it is very well written and explained, and I have only a couple of minor suggestions which are listed below:
Introduction:
Line 104 -106: You stated that “Image-processing analysis of ultrasonograms could potentially be used to predict intramuscular fat content in live beef cattle; however, the accuracy of this method appears to decline with the increasing fat content” and cited the works of Izquierdo et al. 1996. I am a great supporter of Izquierdo et al. work because being pioneers in live beef IMF estimation, however, since 1996 many other attempts have been made, the last one by Fiore et al. in 2020 (Fiore E, Fabbri G, Gallo L, Morgante M, Muraro M, Boso M, Gianesella M. Application of texture analysis of b- mode ultrasound images for the quantification and prediction of intramuscular fat in living beef cattle: A methodological study. Research in Veterinary Science Volume 131, 2020, 254-258) who worked on the same subject achieving higher performances with higher fat contents. When discussing some of the limits of this technique I suggest you consider also the more recent literature for better accuracy and thoroughness.
Discussion:
Line 336-348: Tables 7 and 8 are enclosed here, however, they do not follow the text where they are presented. I suggest moving the tables in the Result section after the paragraph where they are introduced (eg. Table 7 is mentioned in line 305, therefore could be moved to line 316, at the end of the paragraph. Same for table 8, introduced in line 318 and could be moved to line 323) for better clarity.
Line 417: I suggest including some references to ultrasound applied to mammalian muscle mass too along with testicular tissue composition and mammary gland since muscular components are the focus of the study. Among the works published on beef since the year 2000 I can suggest you:
- Aass, L.; Gresham, J.; Klemetsdal, G. Prediction of intramuscular fat by ultrasound in lean cattle. Livest. Sci. 2006, 101, 228–241.
- Aass, L.; Fristedt, C.-G.; Gresham, J. Ultrasound prediction of intramuscular fat content in lean cattle. Livest. Sci. 2009, 125, 177–186.
- Fabbri, G.; Gianesella, M.; Gallo, L.; Morgante, M.; Contiero, B.; Muraro, M.; Boso, M.; Fiore, E. Application of Ultrasound Images Texture Analysis for the Estimation of Intramuscular Fat Content in the Longissimus Thoracis Muscle of Beef Cattle after Slaughter: A Methodological Study. Animals 2021, 11, 1117.
- Hassen, A.; Wilson, D.E.; Amin, V.R.; Rouse, G.H.; Hays, C.L. Predicting percentage of intramuscular fat using two types of real-time ultrasound equipment. J. Anim. Sci. 2001, 79, 11–18.
These are only for the beef sector, but many others can be found too for the swine, working with different fat percentages.
Author Response

(The authors gave the same response as above.)
